# Evaluation of the External Jugular Vein Overlying the Sternocleidomastoid Muscle as Venous Lymph-Node Flap

**DOI:** 10.3390/jcm11071812

**Published:** 2022-03-25

**Authors:** Lorenz Kadletz-Wanke, Felicitas Oberndorfer, Erik Grabner, Lukas Kenner, Klaus F. Schrögendorfer, Gregor Heiduschka

**Affiliations:** 1Department of Otorhinolaryngology and Head and Neck Surgery, Medical University of Vienna, 1090 Vienna, Austria; erik.grabner@meduniwien.ac.at (E.G.); gregor.heiduschka@medunwien.ac.at (G.H.); 2Institute of Pathology, Medical University of Vienna, 1090 Vienna, Austria; felicitas.oberndorfer@meduniwien.ac.at (F.O.); lukas.kenner@medunwien.ac.at (L.K.); 3Unit of Pathology of Laboratory Animal Pathology, University of Veterinary Medicine, 1090 Vienna, Austria; 4Ludwig Boltzmann Institute for Cancer Research, 1090 Vienna, Austria; 5CBMed Core Lab2, Medical University of Vienna, 1090 Vienna, Austria; 6Department of Plastic and Reconstructive Surgery, University Hospital Sankt Poelten, 3100 St. Pölten, Austria; klaus.schroegendorfer@meduniwien.ac.at

**Keywords:** lymph edema, reconstruction, external jugular vein, venous lymph-node flap, lymphatics

## Abstract

Background: Until recently, vascularized lymph-node flaps were based on arterial and venous donor vessels. Now, venous lymph-node flaps form a novel promising concept in the treatment of advanced-stage lymphedema. In preliminary studies, the external jugular vein has shown promising results as a venous lymph-node flap. However, nothing is known about the number of lymph nodes adjacent to the external jugular vein. Methods: Standardized specimens of the external jugular vein and surrounding fatty tissue directly overlying the sternocleidomastoid muscle were obtained during routine neck dissection. Histologic evaluation was performed in order to evaluate for the presence of lymph nodes within the tissue. Results: A total of 20 specimens were evaluated. There was no vein in 4 of the samples. We found lymph nodes in 9 of the remaining 16 samples. In 7 samples, lymph nodes were absent. Conclusion: Our results suggest that the vein directly overlying the sternocleidomastoid muscle may not be the ideal candidate for a venous lymph-node flap.

## 1. Introduction

Lymphedema is a commonly encountered disease. Current statistics report a prevalence rate of 1.33 per 1000, increasing with age up to 5.4 per 1000 [1]. However, several authors state that the prevalence of lymphedema is highly underestimated due to many unreported cases [2].

Globally, lymphatic filariasis is the most frequent cause of this disease. In total, approximately 40 million people are severely infected and disfigured by lymphatic filariasis in endemic areas [3]. In developed countries, lymphedema is most commonly caused by treatment of malignant diseases. Surgical excision or radiotherapeutic interventions may interrupt and destroy the lymphatic drainage. Breast-cancer-related surgeries are one of the most common iatrogenic causes for lymphedema. For example, Rafn et al. recently published a systematic review and meta-analysis and found that the pooled rate of chronic lymph edema after breast cancer surgery has a pooled rate of 4–6% [4]. 

First-line treatment of lymphedema consists of conservative treatment approaches that include complex decongestive physiotherapy, pneumatic compression, compressive garments, bandaging, heat therapy and exercise [2,5,6,7,8]. 

In addition, surgical procedures have been developed to help patients that are affected by chronic lymph edema. There is a variety of reconstructive techniques that might help to reestablish lymphatic flow in affected extremities. 

One way to overcome lymphatic obstruction is the use of lymphaticovenular anastomosis (LVA). Koshima and colleagues were the first to describe the supermicrosurgical anastomosis of lymphatic vessels of the subdermal system [9]. Surgical interventions such as LVA are sometimes more successful in early-stage disease than in patients who have already developed signs of severe skin changes and fibrosis. 

Another strategy is the transplantation of lymph nodes in the form of vascularized lymph-node transfer (VLNT) [10,11]. The use of VLNT has shown excellent results in the reduction of lymph edema [12,13]. Traditionally, lymph-node-containing tissue is harvested with an arterial and venous donor vessel and is then transferred to the donor site. Despite all advances in surgical techniques, conservative approaches have a high value, and most patients usually still require the wearing of compression gear postoperatively. 

However, lymph nodes are mainly linked to the venous system and it is often not clear whether there is an arterial supply at all. Thus, Visconti and colleagues have recently proposed a new concept: the venous lymph-node flap, a vein and the attached lymph nodes are transplanted [14]. They created an experimental model in rats that showed the reestablishment of lymphatic flow after transplantation of a vein with surrounding tissue. In addition, Visconti and colleagues reported the successful transplantation in two cases of a venous lymph-node flap based on the external jugular vein (EJV) and its surrounding tissue located in the posterior triangle of the neck [15]. The principle of venous flaps was first described in 1981 by Nakayama and colleagues [16]. These flaps are based on a large vein with its adjacent tissue, including lymphatic tissue. However, there is no artery included in these kinds of flaps. The harvested flap is then transplanted in a flow-through manner along a vein or artery at the recipient site. Nicoli and coworkers tried to categorize venous lymph-node flaps into six categories depending on their afferent and efferent vessels (arterial or venous) [17]. To date it is not completely understood how those flaps survive without arterial blood supply. It is hypothesized that lymph nodes and surrounding tissue remain viable by diffusion. Some authors even propose that partial ischemic conditions may help to promote lymphangiogenesis. 

Naturally, the success of venous lymph-node flaps depends on the presence of functioning lymph nodes in close proximity to the vein. 

Arguably, the external jugular vein is an easy-to-harvest flap with little donor-site morbidity. However, little is known about the actual number of lymph nodes around the EJV. The purpose of this study is to establish a standardized size of the EJV lymph-node flap to evaluate the number of lymph nodes therein.

## 2. Materials and Methods

### 2.1. Patients

All patients undergoing neck dissection at the Department of Otorhinolarnygology—Head and Neck Surgery of the General Hospital of Vienna were included in this study after obtaining their approval to participate. The recruitment period lasted from February to August 2017. All patients aged from 18–90 years were included. Patients with prior surgery or radiotherapy of the neck were excluded. Furthermore, patients who have a mental condition rendering the subject unable to understand the nature, scope and possible consequences of the study were excluded, as well as any case where prognosis or therapeutic decision could be influenced by resection of the EJV and surrounding tissue. Basic sociodemographic variables such as age and sex were determined. Moreover, the weight and size of all patients was measured. Subsequently, the body mass index (BMI) was calculated. Approval of the institutional research board was obtained prior to the beginning of the study (EK1916/2016).

### 2.2. Surgical Technique

After the skin incision, which was horizontally placed in a skin crease at the level of the thyroid cartilage, and raising of superior and inferior subplatysmal flaps, the EJV was dissected. The EJV and its surrounding tissue were exposed from the parotid gland, over the sternocleidomastoid muscle and ultimately over its posterior border. To compare the specimen, we decided to standardize the cutout. From the intersection of the EJV and the sternocleidomastoid muscle, we harvested 3 cm of the EJV caudad and included a 1 cm stripe of fatty tissue on either side of the vessel. Since we anticipated some variability of the path of the EJV, we decided that the intersection with the sternocleidomastoid muscle was a reproducible landmark. Furthermore, harvesting the EJV superficially to the sternocleidomastoid muscle would also protect the spinal accessory nerve (Figure 1). The specimen was then sent for histologic evaluation.

### 2.3. Histologic Evaluation and Statistics

After fixation in formalin, specimens were split in their longitudinal diameter and fully embedded in paraffin. Sections were made from the whole specimen and examined by an experienced pathologist (FO). The presence of lymph nodes was examined and the total number in each specimen was counted. Moreover, the length of the EJV and the size and volume of the adipose tissue were measured. Descriptive statistics were used to analyze categorical and metrical variables. The mean, median, minimum and maximum of each metrical variable were assessed.

## 3. Results

### 3.1. Patient Characteristics 

In total, 11 patients were included in this study. All patients were diagnosed with squamous cell carcinoma of the head and neck region and a neck dissection was necessary in order to remove manifested or potential lymph-node metastases. The study was conducted at the department of Otorhinolaryngology of the Medical University of Vienna in the period between February and August 2017. The mean age of our cohort was calculated as 64.45 years (range 51–84 years). In nine cases, neck dissection was performed bilaterally, and therefore it was possible to obtain two specimens per patient. We examined a total of 20 specimens that were obtained during neck dissection of seven male patients and four female patients. Additionally, the BMI of our cohort was evaluated. The mean body mass index (BMI) from male patients was 23.34 kg/m^2^ in comparison to the mean BMI of female patients with 25.08 kg/m^2^. Taken together, the mean BMI was 24.00 kg/m^2^ (Table 1).

### 3.2. Evaluation of Specimens 

We examined a total of 20 specimens from 11 patients. In 9 out of 11 patients, the neck dissection was performed on both sides. In four specimens, no EJV was present. Thus, a total of 16 specimens could be evaluated. The mean length of the resected EJV was 2.7 cm. The maximum length of the vein was 3.5 cm and the minimum length measured was 2.4 cm. The average size of the surrounding fatty tissue of the EJV was calculated as 4.04 cm^3^ (range 0.99–12.6 cm^3^) (Table 2 and Table 3).

Next, the number of lymph nodes was evaluated. It was possible to detect two lymph nodes in one sample. Additionally, eight flaps contained a single lymph node, whereas seven samples did not show lymph nodes in histologic examination. As aforementioned, in four samples no EJV was present, and thus we were not able to harvest a flap for further evaluation.

Since neck dissection was performed bilaterally in 9 out of 11 patients, we were able to compare the occurrence of lymph nodes in EJV flaps intrapatiently. In one patient no EJV was present on both sides. One patient had no EJV on one side, but the vein was present on the other side. However, in this case no lymph nodes were found in this sample. In two patients, we were able to find lymph nodes on one side (1 and 2, respectively), and no lymph nodes on the other side. Moreover, we were able to detect a single lymph node on each side in three patients.

Lymph nodes were only found in samples from male patients. Nevertheless, it has to be considered that only 4 of 11 patients were female. In three specimens from two female patients, there was no EJV existent. In two female patients, surgery was only performed at one side of the neck. We could not find any association between the number of lymph nodes, the size of the EJV or the BMI of patients. 

## 4. Discussion

Venous flaps were first described in the early 1980s [16]. However, the exact physiology of venous flaps still remains unclear. Before the innovative work of Visconti and associates [14], research and clinical work in the surgical treatment of lymphedema focused on VLNT with an arterial and venous donor vessel [18,19]. Visconti et al. established the hypothesis that the lymphatic functional unit consists of lymph nodes, lymphatic vessels and surrounding veins [14,15]. They were able to create a rat model and show the potential benefits in the treatment of lymphedema [14]. The lymph-node flap based on the rat’s EJV was able to drastically reduce edema and restore lymphatic drainage. Moreover, their first reported cases in human patients using the EJV as a graft showed promising results [15]. Two patients were treated for lymphedema using the EJV as the donor vessel. In both patients the EJV was transplanted along the greater saphenous vein. 

The functional unit of the venous lymph-node flap consists of a major superficial vein and the adjacent tissue, including lymph nodes. The EJV is one example that could be used for this kind of flap, and its specific anatomy might be associated with potential benefits in contrast to traditional VLNT. In particular, it might be easier to harvest and transplant than the nearby located submental flap, and might have less donor-site morbidity than flaps harvested from the extremities such as the commonly used groin flap. Although the EJV is a very superficial structure and easy to harvest, its localization in the midst of the neck might result in aesthetically unfavorable scars. 

Several studies have shown that a higher number of lymph nodes is associated with an increased reduction in lymphedema, but also that at least one single lymph node is necessary for a functioning unit [20,21]. Nguyen and colleagues were able to demonstrate that three lymph nodes reduce lymphedema quicker than a single one [21]. However, after 3 months both groups showed the same reduction in lymphedema. In this context, we wanted to evaluate the EJV and its adjacent tissue for the presence of lymph nodes. 

Primarily, we wanted to establish a defined lymph-node flap that is easy to harvest. During a neck dissection, the surgical exposure is obviously greater than during an anticipated incision for a flap harvest. We therefore chose the intersection of the EJV and the sternocleidomastoid muscle to both improve reproducibility and reduce the risk to the spinal-accessory nerve. This might be a different localization than described by Visconti et al. [15], who reported that they have harvested a flap from the supraclavicular region along the EJV. 

Our first result regarding the presence of the EJV was surprising. We were not able to find an EJV in 4 out of 20 specimens. So far, there is only limited information about the variability of the EJV found in the literature. As a matter of fact, the absence of an EJV was only described in case reports [22,23]. During a neck dissection, the EJV is usually not of prime interest to the surgeon, so an absence may actually be considered favorably. As there is only very limited data available on the presence or absence of the EJV, our data needs to be interpreted with caution. In particular, only 20 samples were included in our study, which is the major limitation of this study. This warrants further follow-up studies. However, when planning a venous lymph-node flap based on the EJV, we strongly recommend a preoperative neck ultrasound to evaluate the status of the EJV. This should be performed in any case, and even if it is decided to harvest the vein posterior to the sternocleidomastoid muscle. 

For the design of the venous lymph-node flap, we considered a vessel length of 3 cm as sufficient. Secondly, we assumed that a 1 cm-wide strip of fat on either side of the vein would at least contain one lymph node. Yet, according to our data, our investigation revealed no lymph nodes in nearly half of the samples. In detail, we found no lymph nodes in our female patients. However, it has to be mentioned that only 4 of 11 patients were female and only 3 of them had an EJV, so we do not consider sex as a possible contra-indication for this flap. 

## 5. Conclusions

To summarize, we conclude two main points from our study: Since an EJV is not always present, we strongly recommend performing an ultrasound before considering this kind of flap. Secondly, we do not recommend using the EJV superficial to the sternocleidomastoid muscle, as the number of lymph nodes seems to be limited. 

In our analysis, we did not follow the EJV further down towards the clavicle. In this part of the neck, the EJV potentially crosses over the spinal-accessory nerve. Hence, when raising an EJV-dependent flap, meticulous caution must be taken regarding the nerve. We still consider the venous lymph-node flap to be a promising treatment option in patients with lymphedema. However, great attention is required when designing and raising the flap.

## Figures and Tables

**Figure 1 jcm-11-01812-f001:**
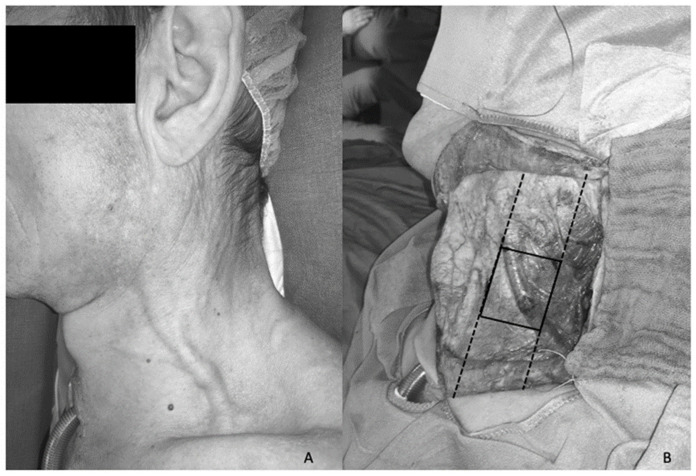
External Jugular Vein. (**A**) Preoperative image of the external jugular vein and (**B**) intraoperative exposure of the vein after raising of the subplatysmal flaps. The dotted lines show the anterior and posterior border of the sternocleidomastoid muscle, and the quadrangle shows the area of resection.

**Table 1 jcm-11-01812-t001:** Basic data of the study population.

	Total	Female	Male
No. patients	*n* = 11	*n* = 4 (36.4%)	*n* = 7 (63.3%)
Neck Dissection			
Unilateral	*n* = 2 (18.2%)	*n* = 1 (9.1%)	*n* = 1 (9.1%)
Bilateral	*n* = 9 (81.8%)	*n* = 3 (27.3%)	*n* = 6 (54.5%)
	Mean		
Age (years)	64.5 (51–84)	65.3 (54–82)	64.1 (81–84)
Body Mass Index (kg/m^2^)	24.0 (16.6–29.7)	25.1 (21.5–29.7)	23.3 (16.6–28.7)

**Table 2 jcm-11-01812-t002:** Evaluation of specimens.

External Jugular Vein		
Present	*n* = 16	80%
Absent	*n* = 4	20%
Number of Lymph nodes		
0	*n* = 7	43.8%
1	*n* = 8	50%
2	*n* = 1	6.2%
	Mean	Range
Length of the Vein	2.7 cm	2.4–3.5 cm
Size of the Sample	4.0 cm^3^	1.0–12.6 cm^3^

**Table 3 jcm-11-01812-t003:** Number of lymph nodes in each specimen.

Patient Number	Left Side	Right Side
1	0	0
2	0	2
3	1	1
4	No Surgery	0
5	No Vein	No Surgery
6	1	0
7	0	1
8	1	1
9	No Vein	0
10	No Vein	No Vein
11	1	1

## Data Availability

Not applicable.

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
