# Peer review of "Evaluation of the External Jugular Vein Overlying the Sternocleidomastoid Muscle as Venous Lymph-Node Flap"

_jcm, 2022, doi:10.3390/jcm11071812_

Round 1

Reviewer 1 Report

The authors describe their results evaluating the lymph-node-containing external jugular vein flap as a future way of treating lymphedema. I congratulate the authors for their work on this interesting topic. While the article is well written and the data is presented clearly, the number of patients is really small for an anatomic study. I would also recommend the authors to change the title of the article in order to underline the fact that they only looked at the external jugular vein over te SCM muscle: Evaluation of the external jugular vein overlying the SCM muscle as venous lymph node flap. As they also state in their discussions, the already described EJV lymph node flap comes from a more caudally position where lymph nodes are more likely to accompany the vein. Therefore, this flap is promising for the treatment of lymphedema. However, ground work still needs to be undertaken in order to establish it in the lymphatic literature.

Author Response

The authors describe their results evaluating the lymph-node-containing external jugular vein flap as a future way of treating lymphedema. I congratulate the authors for their work on this interesting topic.

Thank you very much for taking your time to evaluate our group’s work.

While the article is well written and the data is presented clearly, the number of patients is really small for an anatomic study.

We are aware that the number of patients is rather small. There was a long discussion with our institutional research board about the number of patients included prior to the start of our study. Due to the pilot character of the study their recommendation in order grant approval was that 20 specimens should be included. We tried to highlight the small number and its potential consequences in our discussion.

I would also recommend the authors to change the title of the article in order to underline the fact that they only looked at the external jugular vein over te SCM muscle: Evaluation of the external jugular vein overlying the SCM muscle as venous lymph node flap.  As they also state in their discussions, the already described EJV lymph node flap comes from a more caudally position where lymph nodes are more likely to accompany the vein. Therefore, this flap is promising for the treatment of lymphedema. However, ground work still needs to be undertaken in order to establish it in the lymphatic literature.

Thank you very much. Indeed, it would be more precise to mention the exact donor-site in the title. Therefore, we have changed the title in accordance with you suggestions.

Reviewer 2 Report

Thank you for an interesting article of the venous lymph node flap. A few comments:

  • Please use more up to date prevalence / incidence data then reference 4 which is now 12 years old. I don't think breast cancer related lymphoedema is 49%. You may also want to break that down into arm lymphoedema and breast oedema. 
  •  You state that in severe cases of lymphoedema conservative approach can be limiting and reconstructive techniques are indicated. This is not completely true as often a very early stage of arm lymphoedema is more suitable for LVA surgery for example than a lymphoedema that has developed severe skin changes and fibrotic tissue over time. Please rephrase. Also don't forget that most patients still have to wear compression post surgery and can also remain susceptible to cellulitis lifelong. Conservative approach remains valuable for these lymphoedema patients even around the surgical intervention. 
  • It is a little bit confusing to me how this flap should work exactly. The paragraph is not very clear to me. Lymphnode transfers are getting relatively well established - this vein flap is a different approach? perhaps use drawing or graphic to explain.
  • After the risks of raising the flap are explained and how little lymphnodes there are you still recommend this as a promising flap. Any long term donorsite issues reported ?
